# An Experimental Field Comparison of Wi-Fi HaLow and LoRa for the Smart Grid

**DOI:** 10.3390/s23177409

**Published:** 2023-08-25

**Authors:** Luke Kane, Vicky Liu, Matthew McKague, Geoffrey Walker

**Affiliations:** 1Faculty of Science, Queensland University of Technology, Brisbane, QLD 4000, Australia; v.liu@qut.edu.au (V.L.); matthew.mckague@qut.edu.au (M.M.); 2Cyber Security Cooperative Research Centre, Brisbane, QLD 4000, Australia; 3Faculty of Engineering, Queensland University of Technology, Brisbane, QLD 4000, Australia; geoffrey.walker@qut.edu.au

**Keywords:** Internet of Things (IoT), network architecture, network performance, symmetric key encryption, Wi-Fi, IEEE 802.11ah, Wi-Fi HaLow, LoRa, smart grid

## Abstract

IEEE 802.11ah, or Wi-Fi HaLow, is a long-range Internet of Things (IoT) communication technology with promising performance claims. Being IP-based makes it an attractive prospect when interfacing with existing IP networks. Through real-world performance experiments, this study evaluates the network performance of Wi-Fi HaLow in terms of throughput, latency, and reliability against IEEE 802.11n (Wi-Fi n) and a competing IoT technology LoRa. These experiments are enabled through three proposed network evaluation architectures that facilitate remote control of the devices in a secure manner. The performance of Wi-Fi HaLow is then assessed against the network requirements of various smart grid applications. Wi-Fi HaLow offers promising performance when compared to rival technology LoRa. This study is the first to evaluate Wi-Fi HaLow in an authentic experimental way, providing performance data and insights that are not possible through simulation and modelling alone. This work provides the basis for further evaluation and implementation of this emerging technology.

## 1. Introduction

The smart grid continues to modernise and improve the traditional power grid of the last century through the introduction and use of modern wired and wireless communication technologies. This bi-directional transmission of energy and supporting data are critical to energy security and a complete transition to renewable energy. It enables improvements in areas such as reliability, diagnosis of faults, demand management, energy efficiency, smart metering, and consumer convenience [1]. When discussing such essential critical infrastructure and the integration of data networks into a vital asset such as the electricity grid, network performance and cybersecurity are crucial factors that must be considered.

The recent IEEE 802.11ah standard marketed as Wi-Fi HaLow [2] promises to offer long-range data links and lower power consumption in the sub-GHz bands. There have been numerous studies examining the performance of Wi-Fi HaLow; however, due to the fact that there were no devices in existence, these studies were all theoretical in nature.

This study examines Wi-Fi HaLow in the context of the smart grid. It does this through authentic, real-world testing of the first available Wi-Fi HaLow module, the Silex technology SX-NEWAH based on the NRC7292 chipset from Newracom [3]. The performance of this new module is examined comparatively against IEEE 802.11n (Wi-Fi n) and LoRa, a competing long-range sub-GHz Internet of Things (IoT) communication technology. Our work contributes network architecture designs to support the evaluation of IoT technology securely. The data obtained from these experiments provide valuable insights that are not available through simulated experiments.

### 1.1. Research Scope

Our research aims to validate the performance of Wi-Fi HaLow devices in terms of throughput and reliability at varying physical distances. The performance is compared against Wi-Fi n and LoRa. This is done through a one-to-one connection, with one device acting as a gateway and another acting as a client. The impact that security mechanisms can make on network throughput was also in the scope of this study.

Validating the scalability of the technology was not in the scope of our research project. Network performance was the only consideration for this study. Other performance metrics, such as power consumption, were out of scope.

The maximum coverage distance of LoRa has already been studied at length in previous studies. The maximum coverage range of the performance experiments in this study was 1000 m. This distance is in line with the expected range of Wi-Fi HaLow.

### 1.2. Research Questions and Objectives

This study will address the following research questions:How can we securely conduct authentic performance testing of IoT technologies at varying distances with minimal resources?How does Wi-Fi HaLow perform in a real-world setting, compared with alternative technologies, and do increased security measures impact performance adversely?Is Wi-Fi HaLow a suitable technology choice for smart grid applications?

These research questions will be addressed by realising the following objectives:Design and implement a network performance evaluation architecture for Wi-Fi HaLow, Wi-Fi n, and LoRa.Conduct experiments on the selected technologies to determine their real-world performance in terms of throughput and packet loss across various distances.Examine the requirements of applications in the smart grid and determine if Wi-Fi HaLow meets those requirements.

### 1.3. Our Contribution

The main contributions of our research are:1.The currently available performance figures for Wi-Fi HaLow are theoretical in nature, creating a clear gap in the research. This work provides real-world performance data of the new Wi-Fi HaLow technology obtained using newly available Wi-Fi HaLow hardware. Researchers and network designers can use these data to determine the suitability of Wi-Fi HaLow for various applications.2.Secure network architectures that can be used to evaluate IoT devices so that data can be collected and settings can be remotely changed without needing specialised hardware or software and with minimal human resources.3.An analysis of newer IoT transmission technologies, compared and contrasted with each other and Wi-Fi n, with a focus on Wi-Fi HaLow and its suitability in smart grid applications.

### 1.4. Paper Structure

The remainder of this paper is structured as follows:Section 2: Provides critical background information on Wi-Fi HaLow, LoRa, and the smart grid. A survey of the previous work related to Wi-Fi HaLow performance evaluation will then be presented.Section 3: Presents the three secure network architectures used to conduct this study’s performance measurement experiments. The methodology is outlined, including the hardware and software resources needed to reproduce the experiments.Section 4: Demonstrates the results obtained from each of the conducted performance experiments as they relate to Wi-Fi HaLow, Wi-Fi n, and LoRa.Section 5: Discusses the interesting and insightful observations taken from the performance experiments. Smart grid use cases for the evaluated technology are then discussed.Section 6: Summarises this work’s key findings, the study’s limitations, and some key areas for future work.

## 2. Related Work

This section will first discuss important and relevant background information about Wi-Fi HaLow, LoRa, and the smart grid. Background information will then be provided for the ChaCha20-Poly1305 Authenticated Encryption with Associated Data (AEAD) algorithm, which is used later in this work. Previous work concerned with the performance evaluation and analysis of Wi-Fi HaLow will then be outlined.

### 2.1. Wi-Fi HaLow

IEEE 802.11ah is a new Wi-Fi technology also known as Wi-Fi HaLow. It operates in the unlicensed sub-GHz frequency band with a configurable channel width of between 1 and 16 MHz [4]. The standard requires that all devices support 1 MHz and 2 MHz channel widths at a minimum, with support for 4, 8, and 16 MHz channel widths optional [5]. It operates in a star topology with a predicted coverage range of around 1 km and data rates of up to 78 Mbps [4]. It can theoretically support up to 8192 devices per access point. This fact and its long range make Wi-Fi HaLow an ideal candidate to underpin IoT networks such as smart metering in the Neighbourhood Area Network [4].

With a large number of supported devices per access point, mechanisms must be included to improve media contention and network scalability resulting in fewer potential collisions. This is achieved through improved media access control methods. Wi-Fi HaLow consists of a mechanism called Restricted Access Window (RAW). This mechanism allows restricted access to uplink transmissions by creating time slots for various groups of nodes to transmit [6].

Short-range Wi-Fi solutions are not typically suitable for IoT deployment due to high power consumption. Wi-Fi HaLow includes a mechanism to increase the sleep time of client devices known as Target Wake Time (TWT). TWT allows the client to know during which time period they will have access to the media. This allows the client to sleep until that time occurs, resulting in a reduction in power consumption [7].

Due to its use in IoT networks, Wi-Fi HaLow has had to become more efficient than its standard Wi-Fi counterparts. As a result, media access control headers have been shortened to a maximum length of 24 bytes by removing header fields such as the ID, quality of service, and high throughput fields [4,7]. The data that were previously included in these fields have been relocated to the physical layer [7]. In addition, it also has improvements that allow faster pairing between devices and access points through two mechanisms [4]:1.Centralised Authentication Control—The number of end devices that can send authentication requests at any one time is limited. The access point broadcasts a message that contains a threshold value. When end devices want to authenticate with the access point, they generate a random number. If this number is smaller than the threshold value received from the access point, it will wait until the next broadcast message is received.2.Distributed Authentication Control—The access point maintains various beacon intervals that contain numerous control slots. An end device can randomly select a beacon and slot. If the end device is unsuccessful in its attempt to authenticate, it will attempt authentication later using a different beacon interval and slot.

In addition to adjusting the channel width, the performance of Wi-Fi HaLow can be further tuned through the use of varying Modulation Coding Schemes (MCS). These schemes (MCS0–MCS10) alter the modulation used and the code rate. Altering the MCS can change the expected throughput and the amount of redundant data that is sent to increase network resilience [8].

Wi-Fi HaLow has many potential use cases. Some use cases identified in the literature include implementation in the smart grid, in wireless sensor networks, smart agriculture, and as a way to extend existing wireless networks [9].

Wi-Fi HaLow is compatible with the latest security scheme for Wi-Fi, known as Wi-Fi Protected Access 3 (WPA3). WPA3 provides a greater level of protection against brute force attacks. It does this through a mechanism known as the Simultaneous Authentication of Equals (SAE) [10]. This involves a four-way exchange of authentication frames based on the dragonfly key exchange method [11]. In addition, open networks also benefit from the protection of confidentiality [12] through an enhanced version of open Wi-Fi networks called Opportunistic Wireless Encryption (OWE) [10]. Instead of data being exchanged between the station (client device) and the access point unencrypted, OWE allows message confidentiality based on short-term keys, which are derived from the public keys of the station and access point by way of the Elliptic Curve Diffie–Hellman (ECDH) algorithm [10].

### 2.2. LoRa

LoRa is a proprietary modulation technology developed and maintained by Semtech [13]. As with Wi-Fi HaLow, it operates in the sub-GHz unlicensed band. LoRa falls into the Low Power Wide Area Network (LP-WAN) category, transmission technologies that can operate over large distances with minimal power requirements [14]. LoRa has a maximum ideal coverage range of around 10 km [15,16,17]. LoRa operates using a Chirp Spread Spectrum (CSS) modulation and Frequency Shift Keying (FSK) [18]. This CSS enables LoRa to function over large distances and be resilient to noise and interference [19].

The network performance of LoRa can be customised by tuning a set of parameters to balance the needs of a given use case. These parameters can be altered to affect throughput, provide increased reliability, and improve the coverage range [20]. The main parameters that can be adjusted are Bandwidth (BW), Spreading Factor (SF), and Code Rate (CR). A lower spreading factor will provide faster throughput at the expense of maximum transmission range, with a higher bandwidth having the same effect [21,22]. Modifying the code rate can provide extra redundancy in the transmitted data to improve the resilience of the transmission while reducing throughput [21].

Due to the large coverage range and customisable configuration options, LoRa has found many use cases. Sendra et al. [23] proposed a system to monitor water quality to detect polluted water caused by agricultural run-off. They implemented a prototype network and conducted real-world performance experiments to evaluate the network performance. Singh et al. [24] proposed a LoRa-based system to monitor soil quality and weather in a precision agriculture setting. Through a real-world implementation, they demonstrated a more cost-effective solution compared to current commercially available products. Odongo et al. [25] proposed a smart grid fault monitoring system based on LoRa. In their work, they implemented their proposed system on a real power network and demonstrated its successful application.

LoRa is commonly paired with a protocol called LoRaWAN. LoRaWAN is a media access control protocol. It operates in a star topology and relies on various servers to process data and convert the traffic into IP-based traffic for further use and processing [26]. LoRaWAN is not in scope for this study.

More recently, LoRa has now been made available in a 2.4 GHz version. It is based on the same modulation scheme as its sub-GHz predecessor, so it maintains its resilience to interference and noise [27]. Being a 2.4 GHz technology, it is not subject to duty cycle limitations that can impact the frequency of transmission with LoRa. It can operate with a higher maximum throughput of around 250 Kbps [28]. This study will include only the sub-GHz version of LoRa, as it is a more comparable technology to Wi-Fi HaLow, both operating in the sub-GHz spectrum.

### 2.3. Smart Grid

The smart grid enables the bi-directional transmission of both energy and data to more efficiently and effectively monitor faults, manage demand, improve energy efficiency, and allow various forms of automation such as smart metering applications [1]. The smart grid also allows distributed energy resources such as solar panels and battery systems to be integrated into existing power systems [29].

The smart grid communication networks can be divided into three more specific domains [30]:1.Home Area Network (HAN): The network that is used to manage and control smart devices within the home. These smart devices may also interface with the smart meter to enable more accurate and meaningful usage data. These networks are typically wireless in nature.2.Neighbourhood Area Network (NAN): This network is commonly formed by a group of smart meters within a close geographical area. This network transmits metering data and control information to and from the provider. These networks are typically wireless but can be wired.3.Wide Area Network (WAN): The WAN provides the connection from the HAN and NAN to the provider. The data transferred through the WAN can include the metering data from the NAN and monitoring and control data from substations. This network also facilitates the stability and functionality of the wider power grid. These networks have high bandwidth and low latency requirements, with wired solutions being favoured.

### 2.4. ChaCha20-Poly1305 AEAD

This section will provide background information on the ChaCha20-Poly1305 AEAD algorithm. This algorithm was used to provide data confidentiality and integrity to the prototype LoRa network later in this paper.

ChaCha is a 256-bit stream cipher first proposed by Daniel Bernstein in 2008. It is a modification of his previous Salsa cipher designed to provide better performance and greater resistance to cryptanalysis [31]. The algorithm is typically referred to as ChaCha8, ChaCha12, or ChaCha20, with the number denoting how many rounds are performed in the execution of the algorithm. A single round of ChaCha consists of 48 operations. These operations include 16 additions, 16 XORs, and 16 constant-distance rotations using 32-bit words [31].

Poly1305, similar to ChaCha, was also created by Daniel Bernstein. It was initially proposed as a message authentication code that could be used with the Advanced Encryption Standard (AES) cipher and was named Poly1305-AES. Poly1305-AES works by computing a 16-byte authentication tag of a given message using a symmetric key known to the message sender and receiver [32]. It uses a 16-byte nonce, which can be known but must never be reused under any circumstances [32].

In 2016 Langley et al. from the Internet Engineering Task Force (IETF) proposed the ChaCha20-Poly1305 cipher suite to be used in the Transport Layer Security (TLS) and Datagram Transport Layer Security (DTLS) protocols [33]. This proposal was supported by Google, who had previously integrated this cipher into TLS in the Chrome browser [34]. ChaCha20-Poly1305 proved to be faster on resource-constrained devices than AES-GCM [34].

In our previous research, the speed and efficiency of ChaCha20 has been demonstrated to exceed the performance of AES on multiple resource-constrained devices [35]. Previously, we created an authentication protocol based on ChaCha20-Poly1305 [36]. That work also examined the network performance of LoRa 2.4 GHz networks using our proposed AEAD scheme. The study concluded that minimal additional latency was introduced compared to unencrypted network traffic.

### 2.5. Wi-Fi HaLow Performance Evaluations

Verhoeven et al. [37] conducted a comparative performance simulation study of Wi-Fi HaLow, LoRaWAN, and NB-IoT (Narrowband IoT). They used the ns-3 network simulator to compare the performance of these technologies in a variety of smart cities and smart agriculture applications. In addition, they compared several metrics relating to reliability. The overall findings from the study are that each of the three technologies has different strengths. Wi-Fi HaLow was found to offer the best throughput, NB-IoT was found to be the most reliable, and LoRaWAN provided the best coverage.

Taramit et al. [38] evaluated the network performance of Wi-Fi HaLow-based networks with a particular focus on the RAW mechanism. They created mathematical models to assess the effect that different factors, such as the number of clients and the contention time, had on the network performance. Through their extensive simulations, they determined that it is critical to consider various factors, such as the location of the stations, the density, and the number of available RAW slots. Their work provides others with knowledge to create scheduling algorithms for Wi-Fi HaLow networks. This study focuses on a specific area of the Wi-Fi HaLow network, the RAW mechanism.

Similarly, the research conducted by Taramit et al. [39] also examines channel access related to the RAW mechanism. They proposed a channel allocation scheme that they validated through network simulations. Numerous other studies also focused primarily on the RAW mechanism [40].

Šljivo et al. [41] examined the RAW mechanism and the Traffic Indication Map (TIM) segmentation features of Wi-Fi HaLow to determine if they had an impact on performance for TCP/IP traffic. They specifically examined the effect on scalability energy usage and latency. They evaluated the performance during video streaming scenarios using several IP-based cameras. Through ns-3-based simulations, they determined that Wi-Fi HaLow is a feasible technology for video streaming, with an upper limit of 20 cameras for low-quality video streams.

Aleksandrovs-Moisejs et al. [42] evaluated Wi-Fi HaLow using a combination of virtual machines, the ns-3 network simulator, and Linux docker containers. They devised experiments to measure the number of client stations the network could optimally handle. They were able to successfully simulate 256 stations at a bandwidth of 1 MHz and 2 MHz. The maximum throughput achieved with these 256 stations was 581 Kbps. While this work considers the scalability of Wi-Fi HaLow, it is limited by the simulation platform and hardware constraints.

Brenes and Marín-Raventós [43] evaluated various wireless networking technologies in the context of smart agriculture. They compared various IoT networking technologies such as LoRa, Sigfox, Wi-Fi HaLow, and LTE-M (Long Term Evolution for Machines). Through their evaluation of the features and benefits of each technology, they concluded there is no one best technology, and multi-technology solutions are needed to fulfil the needs of smart agriculture projects. Furthermore, they expressed a need to evaluate Wi-Fi HaLow due to its compatibility with other IP-based technologies leading to a simplified network architecture. Our work builds upon Brenes and Marín-Raventós’ work by providing a real-world evaluation of the Wi-Fi HaLow technology.

Badihi et al. [44] conducted a review of Wi-Fi HaLow. They modelled the network performance, in particular, the downlink performance of the UDP protocol at a bandwidth of 1 MHz and 2 MHz. They found that Wi-Fi HaLow is a suitable choice for controlling IoT actuator devices. They also found that lower latency communications can come at the expense of power consumption. Our study builds upon this work by also evaluating the performance of Wi-Fi HaLow using UDP communications through genuine performance measurement experiments.

Ahmed et al. [45] conducted a study that compared the features, benefits, and performance of Wi-Fi HaLow to IEEE 802.15.4. They concluded that Wi-Fi HaLow is favourable to IEEE 802.15.4 in terms of coverage, throughput, and association time. However, they also determined that IEEE 802.15.4 is more conservative with power consumption under high network traffic conditions. Their results were obtained through various simulations.

Domazetović et al. [8] modelled the performance of Wi-Fi HaLow devices to determine the transmission range that the technology can provide. Through their various mathematical models, they determined that using the MCS10 scheme, a maximum possible range of approximately 1.5 km could be achieved.

Soares and Carvalho [46], through modelling, analysed the performance of the Wi-Fi HaLow network. They examined the throughput across RAW slots. They compared the results of the model with those from an ns-3 simulation and found their mathematical model produced results consistent with the simulation.

There is a clear gap in the research, with theoretical and simulation-based approaches being the favoured research methods. Whilst these studies contribute to the research in the Wi-Fi HaLow space, it is equally important to have studies that address the actual performance of Wi-Fi HaLow. Our research addresses this gap in the literature by providing network architectures for real performance evaluation experiments, as well as real results of Wi-Fi HaLow performance, compared and contrasted with Wi-Fi n and LoRa technologies.

## 3. Materials and Methods

Three test bed environments were constructed to evaluate and measure the three transmission technologies’ performance. It was critical that access between the transmitter and receiver devices was maintained remotely so that configuration could be changed between tests. The configuration could be remotely updated, and the devices could be reset if needed. There were situations where the devices were not in range, so a secondary LTE (Long Term Evolution) connection was used with associated security measures. As the requirements and configurations of the devices used in the tests were different, so too were the supporting test bed topologies. This section will discuss the required resources, the test bench setup, and the method followed during the device evaluation process.

### 3.1. Required Resources

Several resources, both hardware and software, were used throughout these experiments. A full list of requirements to evaluate all three technologies is provided in this section.

#### 3.1.1. Hardware Requirements

This is the hardware used to build the test benches used in this paper. This is the total list of components. When constructing a test bench for just one of the technologies discussed in this paper, some components can be omitted.

2 × laptops (one with an Ethernet interface and cable). The laptops used in this study were Dell Latitude 7400 laptops with an Intel Core i7-8665U processor, 16 GB DDR-4 2666 MHz RAM, and 512 GB NVMe SSD.2 × LTE Modem with SIM card and carrier subscription and USB cables2 × Silex SX-NEWAH-EVK-US IEEE 802.11ah Evaluation Kits (includes a Raspberry Pi and the Wi-Fi HaLow chipset, SD cards and drivers).2 × Mini-USB B to USB-A cables to power 802.11ah radios.2 × ESP32-based microcontroller devices with attached SX1276 LoRa sub-GHz chipsets and USB cables.2 × 900 MHz half-wave dipole antennas with a gain of 0 dBd (0 dB over a dipole). These antennas were used with the Wi-Fi HaLow and the LoRa devices.2 × Geecol Realtek 8812BU-based USB Wi-Fi adapters. These adapters are dual-band IEEE 802.11 ac. The adapter was set to IEEE 802.11 b/g/n mode, and the 5 GHz band was disabled. Testing of 5 GHz Wi-Fi was out of the scope of this study. External 2.4 GHz dipole antennas with a 2.8 dBi gain were attached to the adapters. The adapters’ transmission power was left at its default setting of 22 dBm.1 × portable power bank with USB-C to USB-C cable

The specific hardware requirements as they pertain to each of the three tested technologies can be seen in the architecture diagrams:Wi-Fi HaLow: Section 3.4.Wi-Fi n: Section 3.6.LoRa: Section 3.8.

#### 3.1.2. Software Requirements

This software was used to build the test benches used in this paper. As with the hardware, this is the total list of software. Some software may not be required if just one of the test benches is being constructed.

Microsoft Windows 10/11.Raspberry Pi OS 32-bit (preinstalled on the SX-NEWAH-EVK-US).Visual Studio Code v1.81 with the PlatformIO extension.SSH (Secure Shell) client software.OpenVPN clients for Linux and Windows.Cloud services for Virtual Private Network (VPN) deployment (Amazon Web Services was used for this study).

Other essential resources include the SX-NEWAH Host driver specification sheet (available on request from Silex Technology).

### 3.2. Test Bench General Security Concept

This section outlines the general security concepts to ensure secure device management. You can find the specific test bench architecture for each of the three technologies detailed in Section 3.4, Section 3.6 and Section 3.8.

As mentioned previously, remote access needed to be maintained at all times with the access point to facilitate altering the configuration or resetting the devices in case of errors. It is important to consider the security implications of accessing devices across the Internet. It was important to ensure confidentiality so credentials were not compromised and integrity to ensure control was maintained.

Figure 1 shows the flow of data between the user, the end device, and the access point before security measures were implemented. In this configuration, the user is accessing the Access Point and the End Device via Microsoft Remote Desktop Protocol (RDP) or SSH. Whilst this would work for remote access purposes, having these services exposed to the Internet is a security risk.

Figure 2 shows the flow of data between the user, the end device, and the access point with security measures in place. In this configuration, the user still accesses the access point via RDP and SSH. This time, a VPN server is used to create a secure encrypted tunnel over the Internet. Implementing the tunnel allows confidentiality and integrity to be maintained. The user can access the end device directly through a local connection, so it is not required to be a VPN participant. The user and the access point both authenticate to the VPN server using certificates issued to each participant before the network deployment.

### 3.3. Location

The performance measurement experiments were all conducted at the same physical location. A large parkland location, Minnippi Park in Brisbane, Australia, was chosen for the performance experiments. This location provided a mix of direct line of sight and a moderate level of environmental obstructions such as trees. A map of the area can be seen in Figure 3.

### 3.4. Wi-Fi HaLow Test Bench Architecture

The network architecture used for evaluating Wi-Fi HaLow can be seen in Figure 4. The network consists of two SX-NEWAH-EVK-US devices. The device on the left acts as the access point, while the device on the right is in the role of a station. The blue dashed line represents the Wi-Fi HaLow network connection. The access point device was left in a stationary position throughout the testing, while the station device was moved around to points of various distances from the access point. A portable power bank powered the access point device, while the station device was powered from a laptop via a USB connection.

In the interest of efficiency, it was important that control of the access point could be maintained remotely. This was achieved by implementing a secondary LTE network. An LTE modem was connected directly to the access point, and another LTE modem was connected to the laptop. The brown lines represent the connections in this network.

It was important not only to maintain control of the access point remotely but to ensure this was done in a secure manner. Therefore, a virtual network was created to facilitate a secure encrypted connection and to overcome the limitations of Carrier-grade NAT (Network Address Translation) imposed by the LTE network provider. This network consisted of an OpenVPN access server deployed in the cloud. Certificates were issued from the OpenVPN server to the access point and the laptop to allow them to authenticate with the OpenVPN server. An OpenVPN client was installed on both the access point and the computer.

Finally, an SSH server was running on both the access point and the station. In addition, the laptop was running an SSH client to access both of these devices for data collection and control purposes. The connection to the station was achieved through a direct local Ethernet connection between the laptop and the station. The connection to the access point was via the OpenVPN tunnel.

### 3.5. Wi-Fi HaLow Experiments and Methods

Numerous tests were conducted on the Wi-Fi HaLow devices designed to measure the maximum real-world performance that could be expected. The areas of interest of this study were expected throughput, packet loss, and latency. These were measured using a variety of experiments.

The first set of experiments was designed to determine whether there was a significant difference in throughput experienced between WPA3-secured networks vs. open OWE networks. This experiment set was also designed to measure the maximum expected throughput of Wi-Fi HaLow in ideal conditions. The devices were placed in close proximity to each other in an indoor environment. The throughput was then measured using the iperf utility [47], which is a commonly used network testing tool. Iperf has the ability to create TCP or UDP data streams to measure the throughput of a network. UDP packets were used to measure, as UDP has a lower overhead than TCP. This would allow us to measure the maximum expected throughput more accurately. Iperf ran in server mode on the access point and listened for a connection from the station. Three sequences of experiments were conducted, each lasting 60 s in duration. The results were obtained by averaging the results of each of the three experiments.

The second set of experiments was designed to measure the performance that can be expected from Wi-Fi HaLow at varying distances and to see the relationship between throughput and distance. These experiments were conducted with WPA3 security enabled. Similarly to the first set of experiments, iperf was used in the same configuration. The access point device was left in a stationary position, while the station device was moved to different locations of varying distances.

The third set of experiments was designed to measure the packet loss experienced across various physical distances. Iperf was again used in the same configuration as the above tests. In this set of experiments, ICMP (Internet Control Message Protocol) packets were also sent of varying sizes, being 64, 128, and 255 bytes. In total, 3 sets of 20 pings were sent for the ICMP tests, and the failure rate was captured and averaged.

### 3.6. Wi-Fi n Test Bench Architecture

The network architecture used for evaluating Wi-Fi n can be seen in Figure 5. The network consists of two Microsoft Windows-based laptop computers. One machine acted as the access point, while the other served as a station. The access point was left in a fixed position. Each laptop had a USB Wi-Fi adapter and was fitted with an external antenna. The internal Wi-Fi adapter and Bluetooth radios on each laptop were disabled to minimise the risk of signal interference. A channel scan was performed to determine if other surrounding devices may cause interference. Channel 11 was selected for testing, as there was minimal interference detected. The default channel width of 20 MHz was used. Multiple-Input Multiple-Output (MIMO) was not used, as there was only a single antenna for the access point and the station.

Similar to the previous Wi-Fi HaLow test bench, it was important that remote access could be maintained so that relevant settings could be updated or devices could be reset in case of an error. Again, in the same configuration as the HaLow test bench, an LTE modem was used at both the station and the access point to facilitate a secure cloud-based VPN connection. This time, RDP was used as the means to control the access point device remotely. This was to ensure that control could be maintained if the devices were out of range and prevent RDP traffic from using the test network and potentially impacting the experiment results.

### 3.7. Wi-Fi n Experiments and Methods

For the first set of experiments, the throughput was measured with iperf in the same configuration as the HaLow devices. However, as Wi-Fi n is not designed to be used as a long-range protocol, the testing was limited to a smaller range of 50, 100, and 200 m distances.

The second set of experiments was, again, focused on expected packet loss. Similar to Wi-Fi HaLow, ICMP and UDP packet loss were measured using the same tools and methods.

### 3.8. LoRa Test Bench Architecture

The network architecture used for evaluating LoRa can be seen in Figure 6. In this test bench, there are two Semtech SX1276 LoRa transceivers connected to ESP32 microcontrollers. The LoRa SX1276 was set to operate at a frequency of 915 MHz. As with the two Wi-Fi test benches, one LoRa device acts as the access point and remains in a stationary position, while the other device serves as a station and is moved to various distances from the access point for the purposes of the performance experiments. Each LoRa device had an external antenna of the same specification as the Wi-Fi HaLow devices. The LoRa devices were each connected to a Microsoft Windows-based laptop computer.

As in the previous tests with Wi-Fi HaLow and Wi-Fi n, it was important that remote control of the devices could be maintained. Again, the same VPN cloud infrastructure was used. RDP was used to remotely change the LoRa device parameters when required through a code upload from VSCode with PlatformIO extensions. Much of the changing of the parameters was automated where possible.

### 3.9. LoRa Experiments and Methods

With LoRa-based devices, there is an extensive amount of combinations of parameters that can be tuned to manipulate the throughput, coverage, and data integrity. The most commonly used bandwidths (125 kHz, 250 kHz, and 500 kHz) and spreading factors (SF7–SF12) were selected for the experiments. Each of these combinations of spreading factor and bandwidth was combined with four different code rates (4/5, 4/6, 4/7, and 4/8). The same packet sizes used in the ICMP tests for the Wi-Fi devices were used for all the experiments with the LoRa devices. Those sizes were 64, 128, and 255 bytes.

LoRa does not have any security measures in its default state. As we did not want to involve the complexities and potential overhead of LoRaWAN, we chose to adapt a ChaCha20-Poly1305-based AEAD protocol we proposed in our previous research [36]. This protocol relies on a three-way key establishment process based on the ISO/IEC 11770-2 Key Establishment Mechanism 6 [48]. The packet structure used can be seen in Figure 7. This protocol was proposed for LoRa 2.4 GHz-based networks but has been adapted to function on the LoRa sub-GHz platform. As with Wi-Fi HaLow, we measured the throughput that could be achieved by the LoRa device both with and without security before conducting any coverage testing.

The algorithm for sending the plaintext packets can be seen in Algorithm 1. This algorithm builds the LoRa packet and transmits it. It takes the 5-byte device IDs of the sender $D_{1}$ and the receiver $D_{2}$ and a 5-byte command *M* as inputs. With this overhead of 15 bytes, a variable length payload/data field *P* is inserted to ensure the total packet size is 64, 128, or 255 bytes. This information is then passed to the LoRa transmitter $L_{t}$ and sent. This whole process is timed to calculate the throughput.
**Algorithm 1** Process for Sending a Plaintext LoRa Packet**Require:**    $D_{1}$ {The device ID of the source} $D_{2}$ {The device ID of the destination} *M* {Command header field} *P* {A payload of size 240, 113, or 49 bytes} $L_{t}$ {LoRa transmitter object}1: $L_{t}$.transmit($D_{1}\left|\right|D_{2}||M||P$)

The algorithm for sending the encrypted and tagged packets can be seen in Algorithm 2. As in the previous algorithm, it also takes the device IDs $D_{1}$ and $D_{2}$ and the command field *M* as inputs. The algorithm also requires a 256-bit session key *K* and an initialisation vector/nonce *I* of 64 bits. A variable length payload/data field *P* is used to ensure the total packet size of 64, 128, or 255 bytes. The ChaCha20-Poly1305 algorithm *C* is then passed to *K* and *I*. $D_{1}$ and $D_{2}$ are then passed to the *C* as authenticated data. *M* and *P* are converted to ciphertext $C_{t}$ by *C*. *C* then computes an authentication tag. $L_{t}$ is then passed to $D_{1}\left|\right|D_{2}\left||I|\right|C_{t}\left|\right|T$ for transmission. This entire process is timed, and the throughput is calculated.
**Algorithm 2** Encryption and Tagging Process for Sending a LoRa Packet**Require:**    $K\leftarrow {\left\{0,1\right\}}^{256}$ {32-byte session key} $I\leftarrow {\left\{0,1\right\}}^{64}$ {8-byte initialisation vector} $D_{1}$ {The device ID of the source} $D_{2}$ {The device ID of the destination} *M* {Command header field}*P* {A payload of size 216, 89, or 25 bytes} $C_{t}$ {empty variable to hold ciphertext} *C* {A ChaChaPoly1305 object}*T* {Empty variable to hold authentication tag} $L_{t}$ {LoRa transmitter object}1: *C*.clear()2: *C*.setKey() ←K3: *C*.setIV() ←I4: *C*.addAuthData() $\leftarrow D_{1},D_{2}$5: $C_{t}\leftarrow C$.encrypt(M||P)6: T←C.computeTag()7: $L_{t}$.transmit($D_{1}\left|\right|D_{2}\left||I|\right|C_{t}\left|\right|T$)

As LoRa does not have to rely on maintaining a connection with the access point, the approach taken to the experiments slightly differed with LoRa compared to the Wi-Fi technologies. With the first set of experiments, we measured the maximum rate the device could construct the package and then successfully transmit it with no receiving device. This was measured with the AEAD protocol and in plaintext transmission with no security measures.

The packet loss rate was then measured from the fixed transmitter to the mobile receiver. The packet loss of LoRa was measured up to a distance of 1000 m. LoRa has the potential for greater coverage than 1000 m, however, the purpose of this study is to compare LoRa with the new Wi-Fi HaLow technology, not push LoRa to its coverage limits. The maximum range of LoRa has previously been well studied.

## 4. Results

### 4.1. Wi-Fi HaLow

The results in Figure 8 show that Wi-Fi HaLow has a performance disadvantage when using WPA3 compared with OWE. The results show a maximum average throughput of up to 4985 Kbps. While there is a measurable difference in performance, the impact on performance when using WPA3-PSK security measures vs. an open OWE network is negligible. The largest impact is shown in the 2 MHz bandwidth with a 248 Kbps difference in favour of OWE.

The results in Figure 9 show the throughput that was achieved at various distances. The technology can provide coverage up to around 1 km. The results show that the throughput drops significantly after the 200 m mark.

The results in Figure 10 show the measured packet loss. The results show substantial packet loss figures at the 500 m mark, with very significant packet loss at 700 m and 1000 m. This set of tests also captured the latency experienced in milliseconds (ms) with the round-trip of successful pings. The latency results can be seen in Table 1.

### 4.2. Wi-Fi n

The results in Figure 11 show the measured throughput of Wi-Fi n at various distances. The results show a significant drop in throughput between 100 m and 200 m. Wi-Fi n, as a short-range technology, would have minimal practical application beyond the 200 m distance. The packet loss demonstrated by Wi-Fi n was relatively low, even at 200 m, despite the significant reduction in throughput that was observed. The packet loss results can be seen in Table 2.

### 4.3. LoRa

Table 3 shows the throughput of plaintext LoRa and LoRa with the custom AEAD security scheme. The results are shown for each SF, BW, and CR combination with 64, 128, and 255 byte packet sizes.

We have selected one SF (SF7) with a packet size of 64 bytes to display as a graph, as the amount of data present for LoRa is extensive. All other SFs and packet sizes follow a consistent trend when comparing the AEAD protocol with no security measures. As shown in Figure 12, there is no discernible impact on the throughput when comparing the encrypted/authenticated packets and the plain-text packets. This indicates that the LoRa transmitter is the bottleneck, with encryption and tagging having no noticeable impact on performance.

The results from the LoRa packet loss experiments can be seen in Table 4. This table shows the packet loss results at a distance of 1000 m between the access point and the end device. The average packet loss across different packet sizes and distances can be seen in Figure 13. The packet loss demonstrated by LoRa is consistently low, showing the resilience of LoRa, despite the interrupted line of sight by environmental factors. From our testing, the worst packet loss result is at 1000 m with a maximum packet size of 255 bytes. Still, the total packet loss is low at 1.6%.

## 5. Discussion

There were concerns that Wi-Fi HaLow demonstrated a steep and significant performance drop as the distance from the access point increased. In addition, we had concerns that nearby high-voltage transmission lines near the original testing location may cause unforeseen effects. Therefore, to confirm our initial findings, further tests were conducted on the Wi-Fi HaLow technology at a different site. This site had a clear line of sight between all the testing locations and the access point. The map of the second testing location can be seen in Figure 14.

Interestingly, the throughput was inconsistent with the tests conducted at the first location, with some combinations of distance and bandwidth performing worse and others performing better. This could be due to other interfering factors not being present at the first location as the first location is further away from housing. However, what remains consistent is a significant drop in throughput when the distance increases. This can be observed in Figure 15.

An interesting observation with Wi-Fi HaLow is the higher bandwidth does not always translate to a better throughput result. After a distance of 100 m, the 2 MHz bandwidth consistently provides a higher throughput when compared with 4 MHz.

We observed a mixed result when looking at the ICMP vs. UDP packet loss results. The tests using iperf with UDP tend to send as many packets as possible in the given time. The ping packets sent via the ICMP protocol do not saturate the connection to the same degree as the UDP testing. At a bandwidth of 1 MHz, smaller, more infrequent transmissions, such as those generated using ping, seem to perform substantially better when compared with faster-paced UDP transmissions. The results are not as clear at other distances and bandwidths. For example, at a distance of 500 m, UDP performs more reliably than ICMP at a bandwidth of 2 MHz and 4 MHz.

In both variations of Wi-Fi, it is interesting to note that, as demonstrated by the results of this study, low throughput does not necessarily indicate high packet loss. This low throughput and low packet loss could be attributed to Wi-Fi having retransmission capability at layer two, which the experiments would not capture. This is also supported by the sharp increase in latency observed in the Wi-Fi HaLow results when the distance increases to 500 m from the access point.

This work has examined Wi-Fi HaLow in a real-world setting. It is important to reinforce that one of this work’s technical contributions was to highlight the difference between real-world performance and theoretical performance. Table 5 shows the Wi-Fi HaLow 50 m throughput test results. These three tests were all executed with an MCS 7, which uses the 64-QAM modulation scheme. A significant difference is shown between the theoretical data rate and the actual measured performance. In particular, the 4 MHz bandwidth test could only reach 35% of the theoretical performance.

There are many limitations with the currently available Wi-Fi HaLow devices. Wi-Fi HaLow chipsets have only recently become available for purchase. Currently, the bandwidth is limited to a maximum of 4 MHz rather than the maximum of 16 MHz defined in the IEEE 802.11ah standard. As newer devices and chipsets become available, further testing should occur at all bandwidths to examine if the quality of the newer chipsets and drivers increases and what performance can be achieved using higher bandwidths.

This study did not examine the scalability of any of the tested technologies. Whilst LoRa and Wi-Fi n have been available for some time, and scalability has been well studied, Wi-Fi HaLow has not. It would be particularly interesting to see if Wi-Fi HaLow can meet its theoretical support of 8192 devices per access point and what practical effect this would have on network performance. Whilst numerous studies identified in Section 2 did discuss scalability, all were based on simulations.

During our Wi-Fi HaLow experiments, it was noted that the station would often lose the connection to the access point. Often numerous connection attempts would have to be made in order to execute the tests successfully. This warrants further investigation in the future to determine what the expected stability of Wi-Fi HaLow should be and what main factors contribute to this perceived instability. Some potential causes include signal degradation or device/driver instability.

At a distance of 200 m, the limits of Wi-Fi n coverage start to be reached. It is an interesting observation that under the testing conditions, Wi-Fi n still maintains a lead in terms of throughput when compared with Wi-Fi HaLow. Based on the currently available chipsets for Wi-Fi HaLow, it may be prudent to continue to use standard Wi-Fi solutions to cover distances under 200 m so that a higher level of network performance is maintained.

Through our testing, we have demonstrated that current Wi-Fi HaLow solutions have a maximum useful range at around the 1 km mark. LoRa has been shown in various studies [50,51,52] to consistently reach distances far in excess of this. Wi-Fi HaLow has established itself as a more ’medium range’ solution. Our testing showed LoRa can reach speeds of up to 20.4 Kbps. Wi-Fi HaLow at a distance of 1 km in our initial testing performed significantly worse, with a maximum throughput achieved of only 3.55 Kbps. However, we demonstrated a maximum throughput of 141 Kbps with a direct line of sight. This would indicate that LoRa would be the preferred technology choice in less-than-ideal situations at significant distances.

Due to the medium-range nature of the coverage of Wi-Fi HaLow, it could find use within the HAN and NAN in the context of the smart grid. However, Wi-Fi HaLow would not be suitable for WAN applications, as the coverage distance could not support these applications. An evaluation of the suitability of Wi-Fi HaLow, Wi-Fi n, and LoRa in some common HAN and NAN scenarios can be seen in Table 6. Wi-Fi n does not possess the range required for NAN applications. LoRa has potential suitability in both HAN and NAN applications; however, its throughput capabilities are on the lower end of the use case requirements. Wi-Fi HaLow has a potential application in several of these HAN and NAN scenarios; however, it does not offer sufficient levels of network bandwidth to support more demanding applications such as distribution automation.

## 6. Conclusions

This study has presented a real-world analysis of the performance of Wi-Fi HaLow. This has been presented comparatively with competing IoT transmission technologies. The performance was evaluated not only against LoRa but also against Wi-Fi n technology. Furthermore, network architectures were introduced that were used to evaluate these technologies that provided the ability to set up prototype networks for evaluation purposes, enabling secure remote access and management. Through the discussion in this paper, we determined some potential use cases for Wi-Fi HaLow in smart grid applications.

For the most part, Wi-Fi HaLow was found to offer a higher throughput when compared with LoRa. For use cases that require transmission over 1 km with no line of sight, Wi-Fi HaLow has been demonstrated to have shortcomings, with LoRa performing significantly better with a higher degree of reliability.

For both LoRa and Wi-Fi, it has also been demonstrated through this study that implementing security measures such as WPA3 for HaLow and an AEAD scheme for LoRa is easily achieved. While these schemes have a measurable impact on the network’s throughput, the effect is negligible.

This study has also demonstrated that with a coverage distance of 200 m or less, it may be preferable to use standard Wi-Fi solutions instead of Wi-Fi HaLow.

A limitation of this study was that scalability was not considered. Whilst there are claims that Wi-Fi HaLow can support thousands of station devices per access point, this has not been validated in any real-world studies to date. This should be considered in future work. Another limitation was due to only one Wi-Fi HaLow device being commercially available; it is difficult to determine if any shortcomings found in the performance are due to the particular set of devices used or are inherent to the Wi-Fi HaLow technology. This study allowed the devices to choose the most appropriate MCS. In future work, it may be beneficial to measure each MCS independently to see if more optimal performance can be achieved. As more devices become commercially available, this is another area where future work could occur. In the future, Wi-Fi HaLow should also be evaluated for its tolerance to environmental interference.

## Figures and Tables

**Figure 1 sensors-23-07409-f001:**
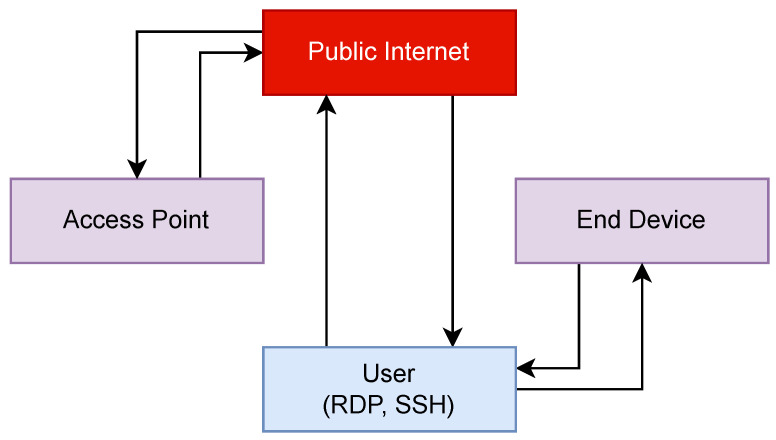
The flow of device management traffic without security measures.

**Figure 2 sensors-23-07409-f002:**
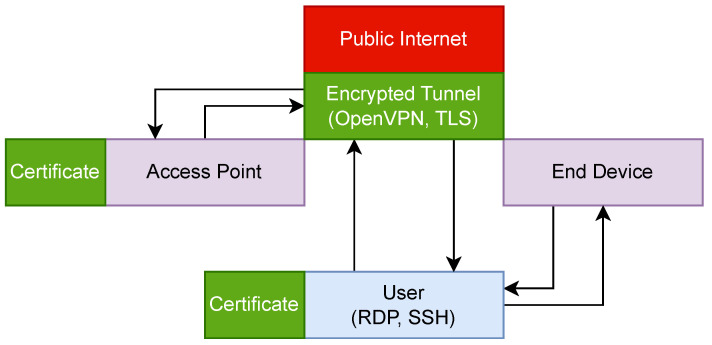
The flow of device management traffic with VPN security measures.

**Figure 3 sensors-23-07409-f003:**
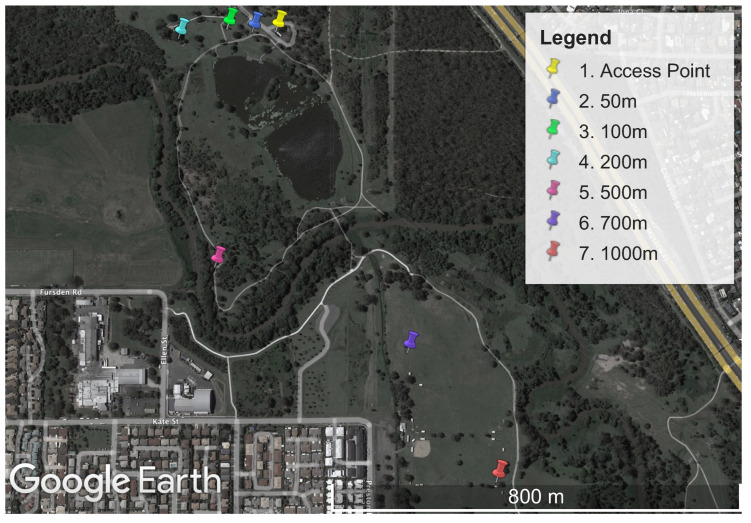
Map of Minnippi Park located in the eastern suburbs of Brisbane, Australia, where the performance experiments were conducted on Wi-Fi HaLow, Wi-Fi n, and LoRa. Each measurement location is plotted as a distance from the stationary access point.

**Figure 4 sensors-23-07409-f004:**
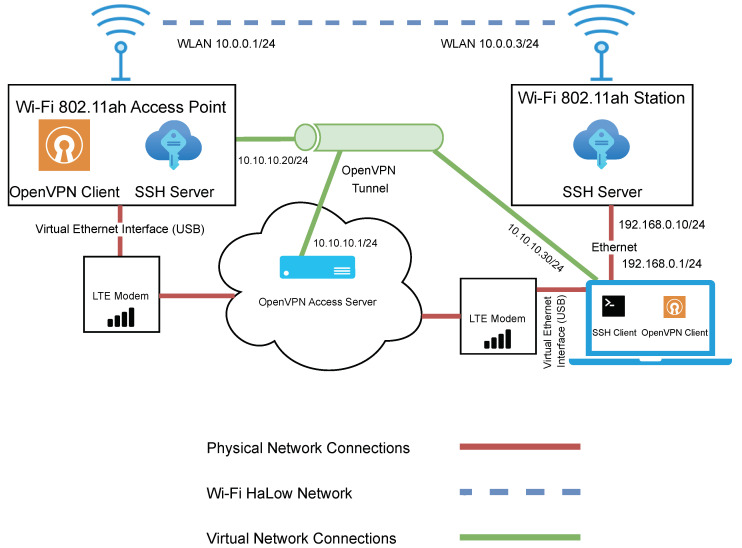
Network architecture diagram for the evaluation of Wi-Fi HaLow.

**Figure 5 sensors-23-07409-f005:**
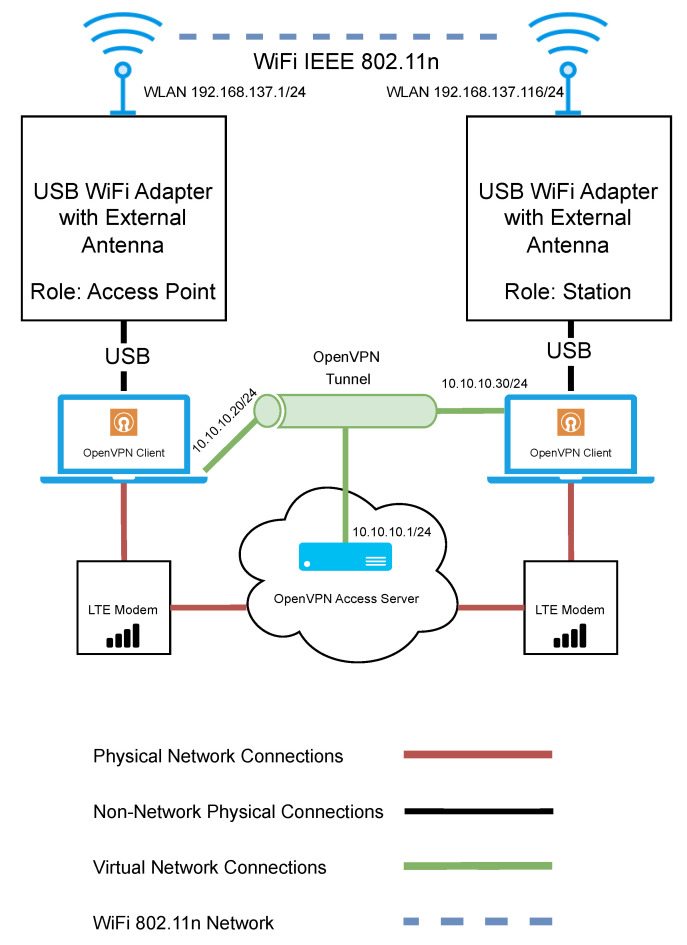
Network architecture diagram for the evaluation of Wi-Fi n.

**Figure 6 sensors-23-07409-f006:**
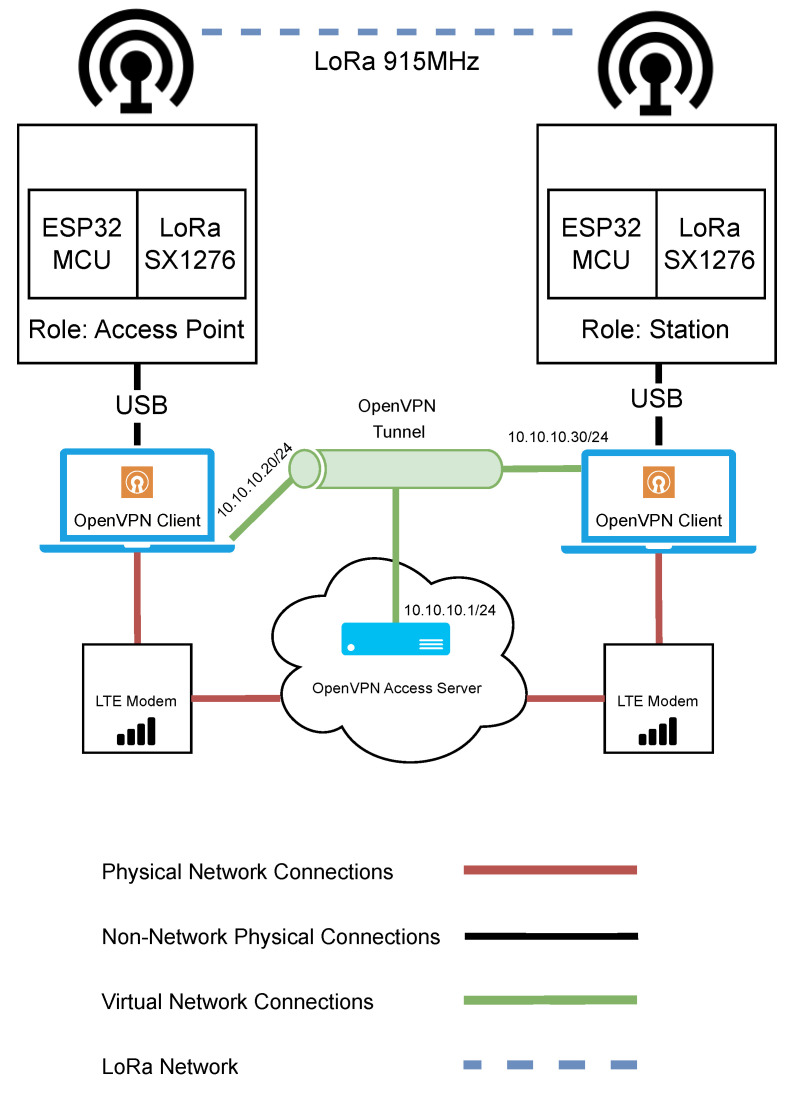
Network architecture diagram for the evaluation of LoRa.

**Figure 7 sensors-23-07409-f007:**
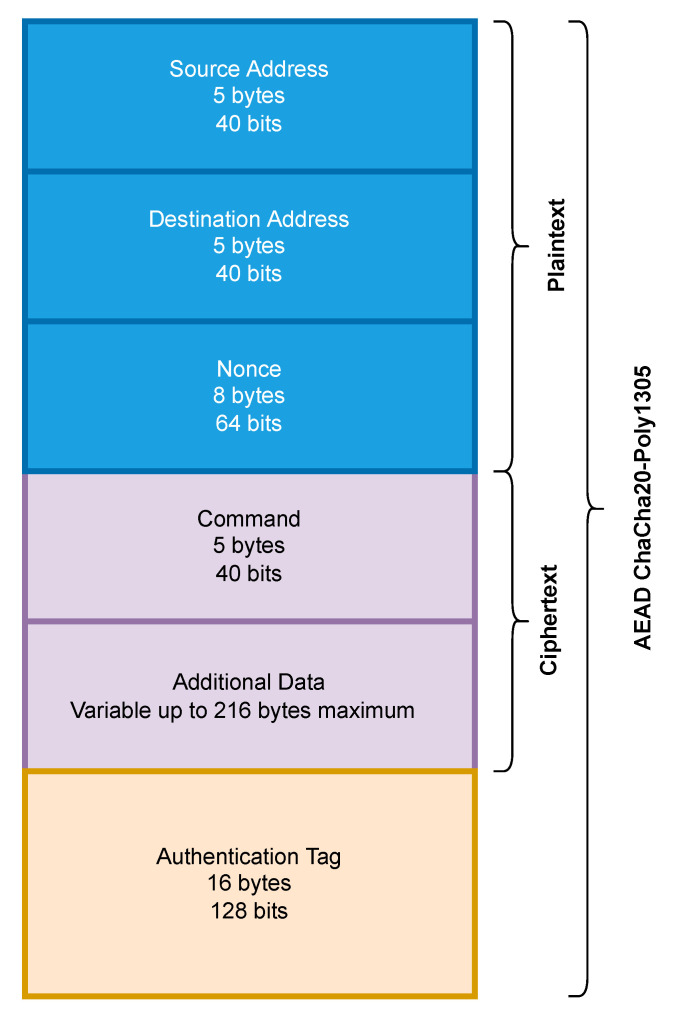
Packet structure adopted from our previous study [36] to support secure LoRa communication.

**Figure 8 sensors-23-07409-f008:**
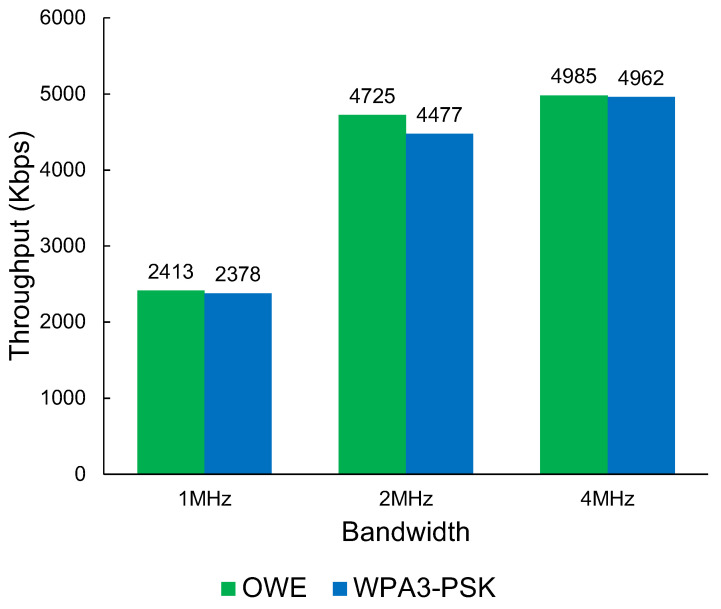
The throughput of Wi-Fi HaLow OWE vs. WPA3-PSK throughput.

**Figure 9 sensors-23-07409-f009:**
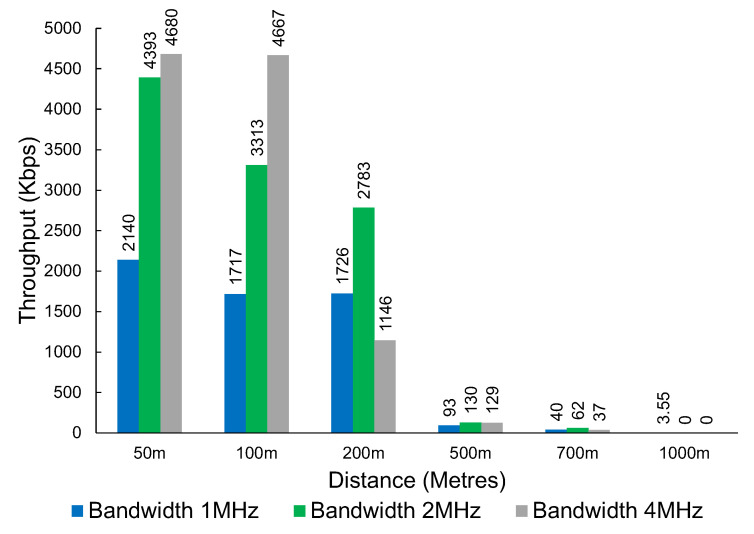
The throughput of Wi-Fi HaLow at various distances and bandwidths.

**Figure 10 sensors-23-07409-f010:**
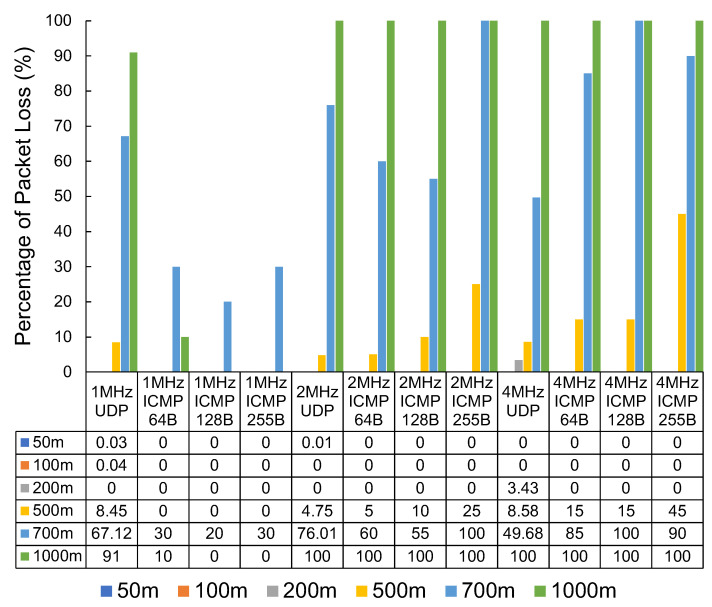
The packet loss percentage of Wi-Fi HaLow by distance, packet type, and bandwidth.

**Figure 11 sensors-23-07409-f011:**
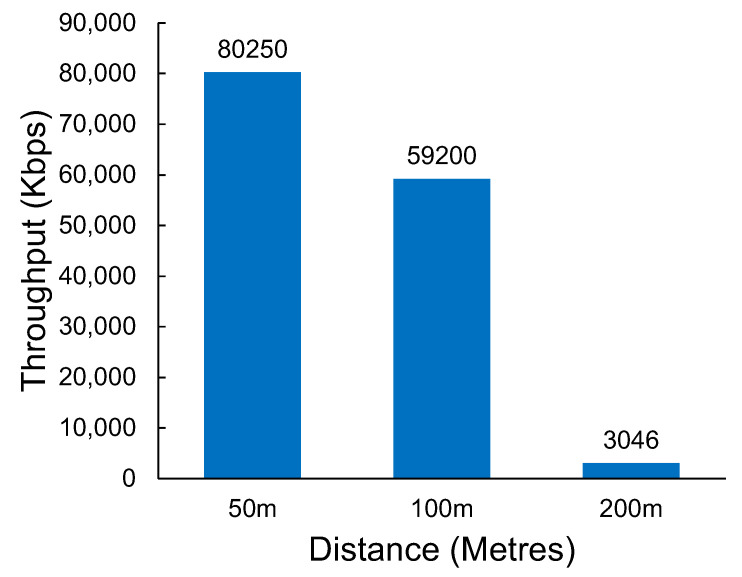
The throughput of Wi-Fi n at various distances.

**Figure 12 sensors-23-07409-f012:**
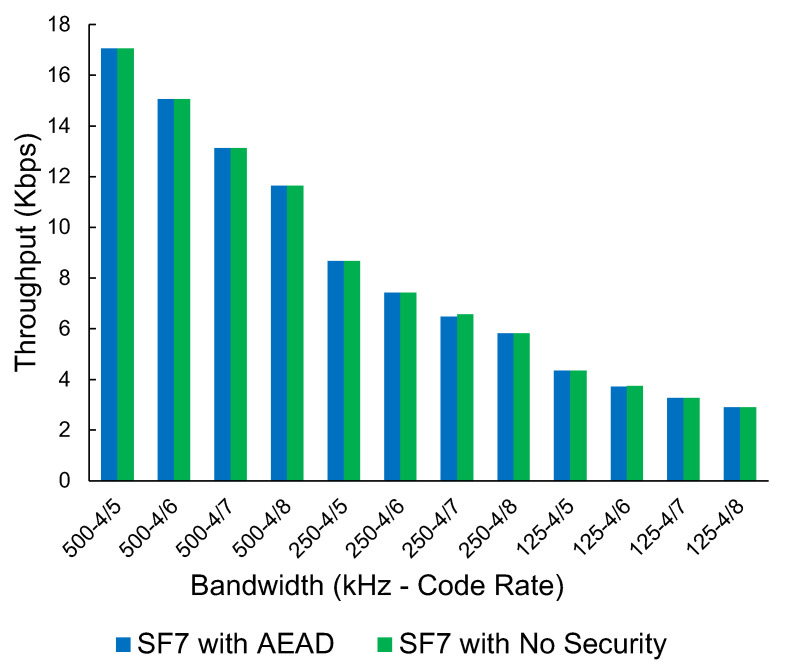
The throughput of LoRa (Kbps), no security vs. ChaCha20-Poly1305-based AEAD scheme at Spreading Factor 7 at various bandwidths and code rates.

**Figure 13 sensors-23-07409-f013:**
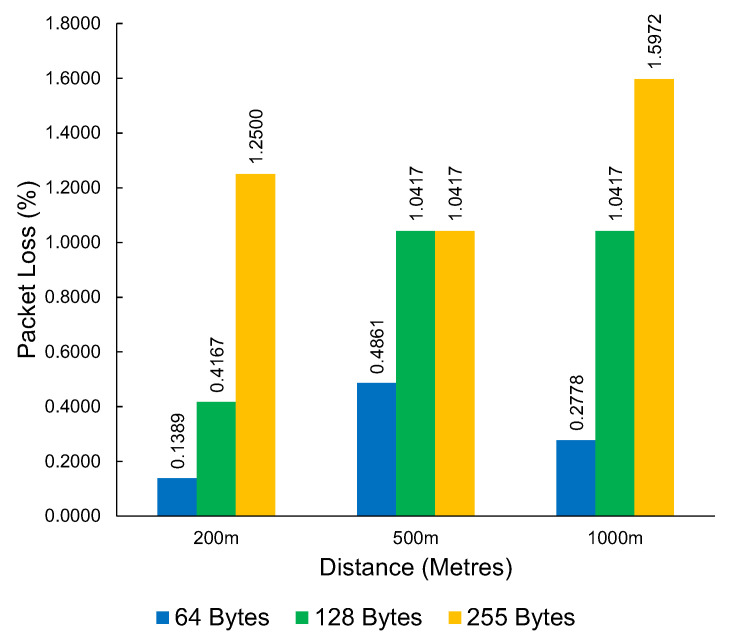
The average packet loss of LoRa at varying distances and packet sizes.

**Figure 14 sensors-23-07409-f014:**
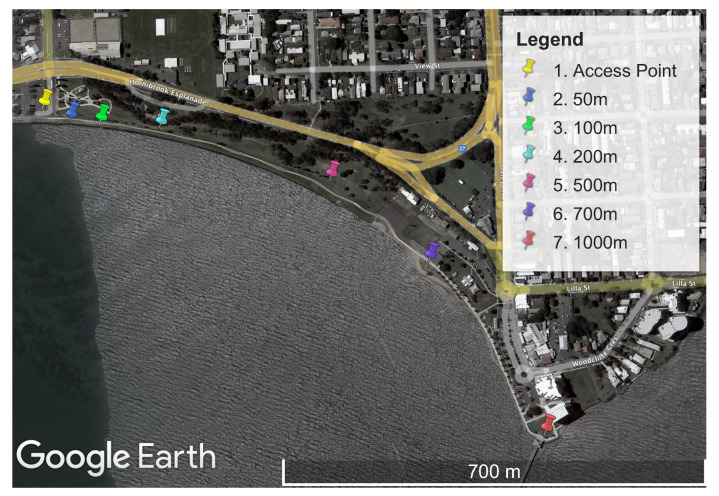
Map of Woody Point, located north of Brisbane, Australia, where the second validating performance experiments were conducted on Wi-Fi HaLow. Each measurement location is plotted as a distance from the stationary access point.

**Figure 15 sensors-23-07409-f015:**
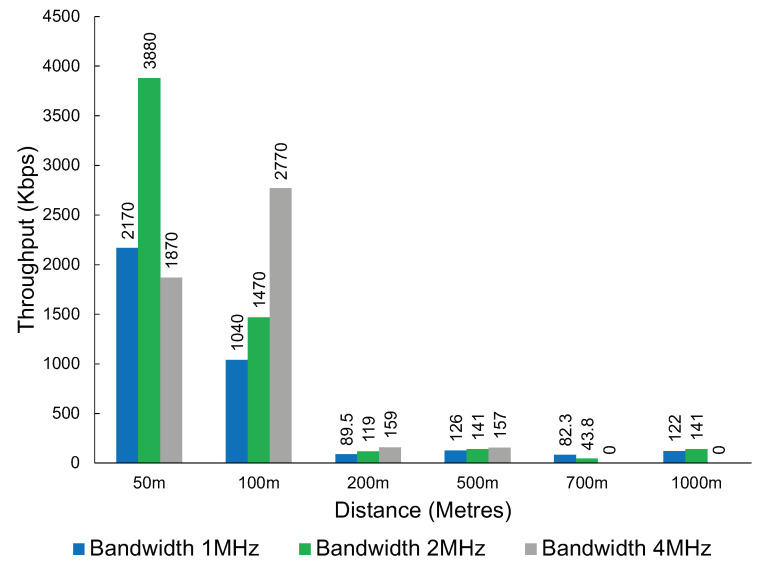
The throughput of Wi-Fi HaLow at a second location was used to verify the significant and sudden performance drop observed in the original performance experiments.

**Table 1 sensors-23-07409-t001:** The latency from successful round-trip pings recorded for Wi-Fi HaLow (ms).

	1 MHz	2 MHz	4 MHz
**50 m**	10.16	8.74	8.38
**100 m**	10.20	8.66	8.66
**200 m**	10.40	11.16	9.50
**500 m**	60.63	56.15	75.52
**700 m**	116.62	96.05	71.98
**1000 m**	95.53	N/A	N/A

**Table 2 sensors-23-07409-t002:** Wi-Fi n packet loss at various distances, protocols, and packet sizes.

	50 m	100 m	200 m
**UDP**	0	0	0.21
**ICMP 64 Bytes**	0	0	10
**ICMP 128 Bytes**	0	0	0
**ICMP 255 Bytes**	0	0	5

**Table 3 sensors-23-07409-t003:** The throughput of LoRa in Kbps AEAD vs. no security measures with all spreading factors and common bandwidth and code rate combinations.

	Packet Size:	64 Bytes		128 Bytes		255 Bytes	
	**Security:**	**AEAD**	**No Sec**	**AEAD**	**No Sec**	**AEAD**	**No Sec**
**SF7**	**500-4/5**	17.067	17.067	18.963	18.963	20.400	20.400
	**500-4/6**	15.059	15.059	16.000	16.000	17.143	17.143
	**500-4/7**	13.128	13.128	13.838	14.027	14.783	14.783
	**500-4/8**	11.636	11.636	12.337	12.337	12.994	12.994
	**250-4/5**	8.678	8.678	9.481	9.481	10.200	10.200
	**250-4/6**	7.420	7.420	8.063	8.063	8.571	8.571
	**250-4/7**	6.481	6.564	6.966	6.966	7.391	7.391
	**250-4/8**	5.818	5.818	6.169	6.169	6.497	6.497
	**125-4/5**	4.339	4.339	4.741	4.763	5.100	5.100
	**125-4/6**	3.710	3.737	4.031	4.031	4.286	4.286
	**125-4/7**	3.261	3.261	3.495	3.495	3.696	3.696
	**125-4/8**	2.909	2.909	3.084	3.084	3.248	3.254
**SF8**	**500-4/5**	9.481	9.481	10.779	10.779	11.525	11.525
	**500-4/6**	8.127	8.127	9.143	9.143	9.668	9.714
	**500-4/7**	7.211	7.211	7.938	7.938	8.361	8.361
	**500-4/8**	6.400	6.400	7.014	7.014	7.365	7.365
	**250-4/5**	4.741	4.741	5.389	5.389	5.763	5.763
	**250-4/6**	4.096	4.096	4.571	4.571	4.846	4.846
	**250-4/7**	3.580	3.580	3.969	3.984	4.189	4.189
	**250-4/8**	3.200	3.200	3.519	3.519	3.682	3.682
	**125-4/5**	2.370	2.370	2.695	2.695	2.881	2.885
	**125-4/6**	2.048	2.048	2.291	2.291	2.426	2.426
	**125-4/7**	1.796	1.796	1.988	1.988	2.094	2.094
	**125-4/8**	1.600	1.600	1.759	1.759	1.843	1.843
**SF9**	**500-4/5**	5.224	5.224	6.059	6.059	6.518	6.518
	**500-4/6**	4.531	4.531	5.146	5.146	5.499	5.499
	**500-4/7**	4.000	4.000	4.472	4.472	4.744	4.744
	**500-4/8**	3.556	3.556	3.954	3.969	4.180	4.180
	**250-4/5**	2.626	2.626	3.021	3.021	3.259	3.259
	**250-4/6**	2.265	2.265	2.573	2.573	2.749	2.749
	**250-4/7**	1.992	1.992	2.241	2.241	2.375	2.375
	**250-4/8**	1.784	1.784	1.981	1.981	2.090	2.090
	**125-4/5**	1.313	1.313	1.513	1.513	1.631	1.631
	**125-4/6**	1.133	1.133	1.286	1.286	1.375	1.375
	**125-4/7**	0.998	0.998	1.119	1.119	1.187	1.187
	**125-4/8**	0.890	0.890	0.991	0.991	1.046	1.046
**SF10**	**500-4/5**	2.926	2.926	3.325	3.325	3.548	3.554
	**500-4/6**	2.547	2.547	2.837	2.837	2.996	2.996
	**500-4/7**	2.246	2.246	2.473	2.473	2.589	2.592
	**500-4/8**	2.008	2.016	2.188	2.188	2.282	2.282
	**250-4/5**	1.467	1.467	1.662	1.662	1.777	1.777
	**250-4/6**	1.270	1.274	1.418	1.418	1.499	1.499
	**250-4/7**	1.123	1.123	1.235	1.235	1.296	1.296
	**250-4/8**	1.006	1.006	1.095	1.095	1.142	1.142
	**125-4/5**	0.734	0.734	0.832	0.832	0.889	0.889
	**125-4/6**	0.636	0.636	0.709	0.709	0.749	0.749
	**125-4/7**	0.562	0.562	0.618	0.618	0.648	0.648
	**125-4/8**	0.503	0.503	0.548	0.548	0.571	0.571
**SF11**	**500-4/5**	1.556	1.556	1.781	1.781	1.950	1.950
	**500-4/6**	1.354	1.354	1.522	1.522	1.646	1.648
	**500-4/7**	1.199	1.199	1.328	1.328	1.426	1.426
	**500-4/8**	1.076	1.076	1.177	1.177	1.256	1.256
	**250-4/5**	0.778	0.779	0.891	0.891	0.975	0.976
	**250-4/6**	0.677	0.677	0.761	0.761	0.824	0.824
	**250-4/7**	0.600	0.600	0.664	0.664	0.713	0.713
	**250-4/8**	0.538	0.538	0.589	0.589	0.628	0.628
	**125-4/5**	0.328	0.328	0.378	0.378	0.408	0.408
	**125-4/6**	0.283	0.283	0.322	0.322	0.344	0.344
	**125-4/7**	0.250	0.250	0.280	0.280	0.297	0.297
	**125-4/8**	0.223	0.223	0.248	0.248	0.261	0.261
**SF12**	**500-4/5**	0.830	0.831	0.960	0.960	1.058	1.058
	**500-4/6**	0.724	0.724	0.821	0.821	0.895	0.895
	**500-4/7**	0.642	0.642	0.717	0.717	0.775	0.775
	**500-4/8**	0.577	0.577	0.637	0.637	0.683	0.684
	**250-4/5**	0.366	0.366	0.416	0.416	0.452	0.452
	**250-4/6**	0.318	0.318	0.355	0.355	0.382	0.382
	**250-4/7**	0.281	0.281	0.309	0.309	0.330	0.330
	**250-4/8**	0.251	0.251	0.274	0.274	0.291	0.291
	**125-4/5**	0.183	0.183	0.208	0.208	0.226	0.226
	**125-4/6**	0.159	0.159	0.177	0.177	0.191	0.191
	**125-4/7**	0.140	0.140	0.154	0.154	0.165	0.165
	**125-4/8**	0.126	0.126	0.137	0.137	0.145	0.145

**Table 4 sensors-23-07409-t004:** Packet loss percentage of LoRa at a distance of 1000 m with all commonly used bandwidths, spreading factors, and code rates.

		SF7	SF8	SF9	SF10	SF11	SF12
**64 Bytes**	**125-4/5**	0	0	0	0	0	0
	**125-4/6**	0	0	0	0	0	0
	**125-4/7**	0	0	0	5	0	0
	**125-4/8**	0	0	0	0	0	0
	**250-4/5**	0	0	0	0	0	5
	**250-4/6**	0	0	0	0	0	0
	**250-4/7**	0	5	0	0	0	0
	**250-4/8**	0	0	0	0	0	0
	**500-4/5**	0	0	5	0	0	0
	**500-4/6**	0	0	0	0	0	0
	**500-4/7**	0	0	0	0	0	0
	**500-4/8**	0	0	0	0	0	0
**128 Bytes**	**125-4/5**	5	0	0	5	0	0
	**125-4/6**	0	0	5	5	0	0
	**125-4/7**	0	0	0	5	0	0
	**125-4/8**	0	0	0	5	0	0
	**250-4/5**	0	0	0	0	10	0
	**250-4/6**	0	0	0	0	5	0
	**250-4/7**	5	0	0	0	5	0
	**250-4/8**	0	0	0	0	5	0
	**500-4/5**	0	0	0	0	0	5
	**500-4/6**	0	0	0	0	0	5
	**500-4/7**	0	0	0	0	0	0
	**500-4/8**	0	0	0	0	0	5
**255 Bytes**	**125-4/5**	0	0	5	10	0	0
	**125-4/6**	15	0	5	5	0	0
	**125-4/7**	0	0	5	5	0	0
	**125-4/8**	0	0	0	5	0	0
	**250-4/5**	0	0	0	5	5	0
	**250-4/6**	0	0	0	5	5	0
	**250-4/7**	0	0	0	0	10	0
	**250-4/8**	0	0	0	0	5	0
	**500-4/5**	0	0	0	0	5	5
	**500-4/6**	0	0	0	0	0	5
	**500-4/7**	0	0	0	0	0	5
	**500-4/8**	0	0	0	0	0	5

**Table 5 sensors-23-07409-t005:** The actual throughput of Wi-Fi HaLow at a distance of 50 m compared with theoretical throughput.

Bandwidth	MCS	Modulation	Theoretical Data Rate (Kbps) [49]	Actual Data Rate (Kbps)	Actual vs. Theoretical Data Rate %
1 MHz	7	64-QAM	3000	2140	71%
2 MHz	7	64-QAM	6500	4393	68%
4 MHz	7	64-QAM	13,500	4680	35%

**Table 6 sensors-23-07409-t006:** Suitability of the tested technologies for HAN and NAN smart grid applications.

Use Case	Bandwidth	Payload Size	Latency	Wi-Fi HaLow	Wi-Fi n	LoRa
Advanced Metering Infrastructure including Electricity Meter Reading and Pricing Updates [53,54]	10–500 Kbps	100–2400 bytes	15 s–4 h	Suitable	Unsuitable	Suitable
Firmware Updates [53]	Unspecified	400–2000 kB	<2 min–7 days	Suitable	Unsuitable	Suitable
Distribution Automation [53]	100 Kbps–10 Mbps	25–1000 bytes	<4 s	Unsuitable	Unsuitable	Unsuitable
Home Energy Management System/Home Automation [53,54,55]	9.6–56 Kbps	10–100 bytes	2–15 s	Suitable	Suitable	Suitable
Electric Vehicle to Grid and Charging [53,54,55]	9.6–56 Kbps	100–255 bytes	<2 s–5 min	Suitable	Suitable	Suitable

## Data Availability

The data presented in this study are available on request from the corresponding author.

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
