# Peer review of "An Experimental Field Comparison of Wi-Fi HaLow and LoRa for the Smart Grid"

_sensors, 2023, doi:10.3390/s23177409_

Round 1
Reviewer 1 Report
This paper presented a real-world analysis of the performance of Wi-Fi HaLow. I have the following comments:
1. I suggest the authors to add a Structural diagram to show the security enhancement of the network architectures.
2. Some attack-defending strategy based on brute force searching has been proposed in the control theory, such as [Secure state estimation against sparse sensor attacks with adaptive switching mechanism], which should also be discussed in the introduction.
3. In general, the use of encryption or coding mechanism will bring the computational burden such the system performance is damaged. Please give detailed analysis of the computational complexity of the proposed encryption/coding mechanism.
4. The "technical" contributions compared with the existing results should be further highlighted.
By my evalution, the presented English in the current version is generally good.
Reviewer 2 Report
Authors evaluate the network performance of Wi-Fi HaLow in terms of throughput, latency, and reliability against IEEE 802.11n (Wi-Fi n) and a competing IoT technology LoRa.
There are some issues that must be fixed in this paper in order to accept it:
The structure of the paper is not good. Introduction section has too many sections. Some of them are just 1 paragraph.
1.4. Paper Structure must provide the structure of the rest of the paper section by section.
Authors should combie 1.5 and 1.6 in a new section 2, titled "related work".
I find missing the citation of a related work such as:
LoRa-based Network for Water Quality Monitoring in Coastal Areas. Mobile Networks and Applications, DOI: 10.1007/s11036-022-01994-8
In section 2.1.1 should must detail the labotop features in terms of computing capacity, memory, etc.
I suggest to remove the border lines of the graphs from figure 6 to figure 13.
Figure 11 should be better explained.
Round 2
Reviewer 1 Report
I have no further comments.
Reviewer 2 Report
Authors have fixed all my comments. The paper is ready to be published.